# Mimicking or Reasoning: Rethinking Multi-Modal In-Context Learning in Vision-Language Models

## Abstract

Vision-language models (VLMs) are widely assumed to exhibit in-context learning (ICL), a property similar to that of their language-only counterparts. While recent work suggests VLMs can perform multimodal ICL (MM-ICL), studies show they often rely on shallow heuristics such as copying or majority voting, rather than true task understanding. We revisit this assumption by evaluating VLMs under distribution shifts, where support examples come from a dataset different from the query. Surprisingly, performance often degrades with more demonstrations, and models tend to copy answers rather than learn from them. To investigate further, we propose a new MM-ICL with reasoning pipeline that augments each demonstration with a generated rationale alongside the answer. We conduct extensive and comprehensive experiments on both perception- and reasoning-required datasets with open-source VLMs ranging from 3B to 72B and proprietary models such as Gemini 2.0 and 2.5. We conduct controlled studies varying shot count, retrieval method, rationale quality, and distribution. Our results show limited performance sensitivity across these factors, indicating that current VLMs fail to effectively utilize demonstration-level information and thus do not inherit the strong few-shot abilities of large language models (LLMs). We further conduct a mechanistic analysis showing that VLMs exhibit weak prefix matching and lack induction-head-like behavior, which potentially explains the failure of MM-ICL.

## 1 Introduction

Vision-language models (VLMs), inspired by the success of large language models (LLMs), are widely believed to exhibit the ability of in-context learning (ICL), i.e., learning from a few examples provided in the prompt without any parameter updates. This capability has been well-documented in LLMs (Brown et al., 2020; Wei et al., 2022; Dong et al., 2022; Khattab et al., 2023; Zhou et al., 2023; Ge et al., 2025), and recent work suggests that VLMs may inherit similar behavior through large-scale multimodal pretraining (Zong et al., 2024) and are capable of performing multimodal in-context learning (MM-ICL) (Qin et al., 2024; Baldassini et al., 2024; Awadalla et al., 2023; Bai et al., 2023). However, several studies (Baldassini et al., 2024; Qin et al., 2024; Chen et al., 2024b) question whether current VLMs are truly learning from demonstrations. Instead, they find that VLMs often rely on shallow heuristics such as copying recent similar responses or defaulting to majority-vote patterns over the demonstrations-rather than acquiring a deeper understanding of the task.

To further probe this issue, we begin by testing under distribution shift, where support and query examples originate from different datasets. Counterintuitively, we observe that model performance often plateaus or even degrades as the number of shots increases, despite being given more demonstrations. This contrasts with the in-distribution case, where performance reliably improves with more demonstrations. We also find failure cases where the model simply copies answers from the demonstrations, rather than learning from them. These observations raise a central question: *do VLMs truly learn from in-context demonstrations, or are they just matching superficial patterns?*

To explore this question, we propose to evaluate whether VLMs can move beyond surface-level pattern matching and truly learn from in-context demonstrations in a new setting. Rather than providing only final answers, we enrich each demonstration with a detailed **reasoning process** (Jiang et al., 2025; Wang et al., 2025b; Lu et al., 2024; Hao et al., 2025; Gao et al., 2024; Yang et al.,

2024; Štefánik and Kadlčík, 2023), i.e., explicit step-by-step rationales that make the task-solving strategy clear. By increasing the informational content of each example, we aim to give models a stronger learning signal and a better chance to internalize the methodology behind the task, rather than relying on shallow cues. To achieve this, we leverage the capabilities of **VLM Reasoners** (Shen et al., 2025; Xu et al., 2024a; Wang et al., 2025a), which inherently generate rationales and answers simultaneously, to assess whether access to intermediate reasoning steps helps models generalize more effectively from demonstrations. Our contributions are as follows:

**(1)** To the best of our knowledge, this paper is the first to study the MM-ICL of VLMs from the lens of reasoning. Using information-enhanced demonstrations with reasoning components, we benchmark the MM-ICL capability of modern VLMs and reach conclusions with more solid evidence.

**(2)** To fairly evaluate MM-ICL for VLM reasoners, we introduce a new **MM-ICL with Reasoning** pipeline that resolves a key format mismatch in prior work: instead of supplying only answers in demonstrations while expecting rationale-plus-answer outputs, we provide demonstrations with both a Pseudo Reasoning (a generated rationale) and an answer. This consistent support-query format improves performance across models and datasets over inconsistent ones.

**(3)** We conduct extensive controlled studies by varying shot count, retrieval method, rationale quality, and distribution. Our analysis reveals that MM-ICL are largely insensitive to these factors, showing limited performance variation across different configurations. We reveal a counterintuitive failure mode showing that current VLMs do not effectively leverage demonstration-level information, challenging the belief that they inherit few-shot learning abilities from LLMs.

**(4)** We provide an attention-level perspective, showing that VLMs demonstrate weak prefix matching and no clear induction-head–like behavior, potentially explaining their limited MM-ICL performance.

## 2 RELATED WORKS

**Multimodal In-Context Learning** Large VLMs have the emerging ability to answer an unseen question or perform a new task without additional training, a capability known as zero-shot learning. Moreover, researchers have found that these models can often achieve better performance when multiple demonstrations of solutions to similar tasks are presented to the model before querying the question (Brown et al., 2020). LLMs have shown strong ICL abilities—learning from demonstrations without parameter updates (Brown et al., 2020; Wei et al., 2022; Dong et al., 2022). VLMs, built on LLMs and pretrained on large-scale multimodal data, are believed to inherit similar capabilities. Recent benchmarks (Zong et al., 2024) and follow-up studies (Qin et al., 2024; Xu et al., 2024b) have evaluated MM-ICL across tasks and analyzed factors such as retrieval, prompt design, and modality contributions. However, these efforts assume that VLMs are capable of MM-ICL, without first establishing whether models actually understand and learn from demonstrations. This motivates a deeper investigation into what VLMs learn in the MM-ICL setting.

**Vision-Language Reasoning Models** To enhance reasoning in VLMs, recent work has focused on post-training techniques and curated datasets. Reinforcement learning (RL)-based approaches, such as Group Relative Policy Optimization (GRPO), have been applied to improve performance across tasks like referring expression comprehension and open-vocabulary detection (Shen et al., 2025; Wang et al., 2025a). In parallel, non-RL strategies—such as preference optimization (Wang et al., 2024)—have shown success in reducing hallucination and improving multi-step reasoning. Additionally, structured reasoning datasets like LLaVA-CoT (Xu et al., 2024a) offer a complementary path by enabling fine-tuning with explicit reasoning supervision. Together, these advances reflect growing interest in building VLMs that can reason more reliably and systematically.

## 3 RETHINKING THE SUCCESS OF MULTIMODAL IN-CONTEXT LEARNING

### 3.1 A CLASSIFICATION OF MM-ICL TASKS

The tasks used for MM-ICL capability benchmarking can be roughly categorized into two categories, depending on whether the problem is well-defined without demonstrations.

**Case I: well-defined tasks without demos:** Visual Question Answering (based on common sense or factual knowledge), Image Captioning, etc. Tasks with solutions uniquely determined by the query.

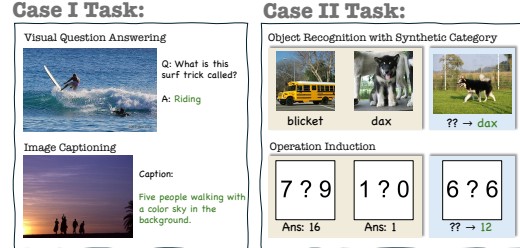

Figure 1: **Left:** Performance difference between ID and OOD using random retriever. **Middle**: Performance of different retrieval methods on OK-VQA. ID: OK-VQA as support set. OOD: TextVQA as support set. We include the unimodal retriever to highlight that the multimodal retriever achieves the best performance in the ID setting, consistent with Qin et al. (2024). **Right**: Wrong answer format directly increases error rate.

**Case II: ill-defined tasks without demos:** Operator Induction, Open-Ended Object Recognition (with synthetic category), etc. Tasks are well-defined only when demonstrations of successfully solved cases are presented (Zong et al., 2024).

We present examples of Case I/II tasks in Fig. 2. While both cases have been studied in the MM-ICL literature, it is less clear how to quantify whether the model succeeds in learning from the demonstration examples for tasks of Case I. Many essential tasks of great practical utility related to perception or reasoning fall in Case I. These tasks have instructions that are clear to understand and follow even when no demonstration is provided, and thus are tractable for VLMs to solve in a zero-shot setting to some

Figure 2: Examples from Case I/II tasks. Case II tasks are ill-defined if no demos are given.

degree. Therefore, it's natural to raise the question: can ICL *truly* enhance a model's capability to solve these tasks as a type of inference-time scaling technique? To answer this question, we investigate how to benchmark and determine *fairly* VLMs' ability to learn from demonstrations in a multimodal scenario, as a primary focus of this work.

## 3.2 A Closer Look at A Performance Gap

To understand whether VLMs can learn from demonstrations, we begin by exposing them to a more realistic setting, where the demonstration data originates from a different dataset with a distinct distribution from that of the queries (Mosbach et al., 2023). Note that this setting closely reflects the real-world user case of ICL, as it's often unrealistic to provide highly relevant demonstrations with ground-truth answers to aid the learning process for entirely new, unseen questions. We denote this setting as **Out-of-Distribution** (OOD), in contrast to the **In-Distribution** (ID) setting where we select demonstrations from the training split of the query dataset. We still enforce that the OOD support set shares the same task type as query data to ensure the evaluation is reasonable.

The primary motivation behind this experiment design is to evaluate whether the model *truly* understands how to perform the target tasks better by mastering the methodology behind them, or it only gains a superficial understanding through memorizing information presented in the demonstrations. To make an analogy, this setting provides the student (VLM) with a test paper (query data) that contains problems not merely variants of the questions (OOD support set) it has seen, but can still be solved using similar methods (same task type).

We consider the task of Visual Question Answering (VQA), a classic problem that falls under Case I discussed above. We use TextVQA (Singh et al., 2019) and OK-VQA (Reichman et al., 2023) and evaluate the performance of OpenFlamingo (Awadalla et al., 2023) and IDEFICS2 (Laurençon et al., 2024). These datasets and models are popular choices for studying MM-ICL in the literature (Baldassini et al., 2024; Qin et al., 2024). Each model will be asked to answer the question in a short response with 1-2 words using the instruction prompt `"Answer the question using a single word or phrase."` whenever necessary. The accuracy of the response is computed through **exactly matching** it to the provided answer candidates. The results are presented in Fig. 1.

We notice an intriguing performance difference between ID and OOD settings (Fig. 1 Left) with the increase number of shots. We observe that the accuracy of OpenFlamingo **monotonically increases**

as the number of shots grows, while such trends are rarely observed in the OOD setting. In the OOD settings, there is a minimal or no increase in accuracy with more demonstrations. This is rather counterintuitive since the model was offered with strictly more correct information (though some of it was less relevant due to OOD); thus, the model should not perform worse than in the zero-shot setting. Additionally, as shown in Fig. 1 (Middle), retrieval-based methods consistently outperform random selection in the ID setting, while the opposite holds in the OOD setting. These observations suggest that the hidden factor that drives the performance gap between ID and OOD cases is not directly relevant to the model's capability to solve such tasks.

One possible factor behind this phenomenon is the response format. When presented with ID demonstrations, the query and support examples share a similar format, making it easier for the model to pick up and follow the expected answer style. However, when presented with numerous OOD examples in distinct formats, the model struggles to generalize the formatting instructions, leading to degraded performance. An example of such an error is presented in Fig. 1 (Right), where a correct answer from OpenFlamingo is deemed as wrong due to a mismatch of answer formats (two words instead of one). A similar conclusion is also mentioned in Zong et al. (2024), where an LLM is used to judge whether a response semantically aligns with the ground-truth answer, instead of relying on an exact-matching function. Compared to exact-matching, LLM judges are less sensitive to answer formats and therefore are more robust for accuracy evaluation. Using LLM judges would drastically reduce the performance gain margin over zero-shot results, supporting our observation that the VLMs like OpenFlamingo **might not learn anything more than formatting** from the demonstrations.

We also present examples of success and failure of MM-ICL in Fig. 3. In the presented cases, when the query is spuriously correlated with image-question pairs in demonstrations, the model latches onto the superficial similarity and directly copies the support answer. In contrast, when an irrelevant example is included, it disrupts this superficial

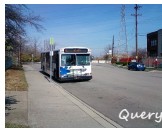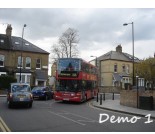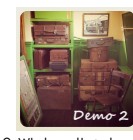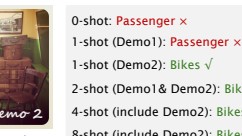

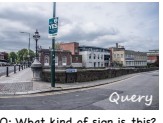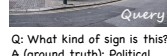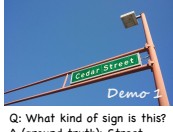

Figure 3: Success and failure of MM-ICL with IDEFICS2.

pattern matching, preventing the model from copying and exposing its inability to truly "learn from" relevant demonstrations to answer the query.

## 4 MM-ICL WITH REASONING FOR VISION-LANGUAGE MODELS

While the results presented in Sec. 3.2 are surprising, these experiments would not be enough to claim anything about whether large VLMs have the *true* ICL capability for Case I tasks for the following reasons. First, despite being widely selected as benchmarks for MM-ICL tasks due to their decent performance, OpenFlamingo and IDEFICIS2 no longer represent the state-of-the-art VLMs, thus it would be biased to draw any conclusions just based on their evaluation performances. Furthermore, except for the question-answer pair, very limited information is provided during the ICL stage, which potentially prevents VLMs from extracting deeper, more meaningful information beyond the answer format and further restricts the performance gain from an increased number of shots.

To address these issues, we propose to benchmark the MM-ICL ability of **modern VLMs** in a new setting, where we provide the model with information-enriched demonstrations to maximize the utility of each example. We achieve such information augmentation by introducing **reasoning process** into the demo, and each presented example would contain a detailed step-by-step thinking process instead of a single answer. By doing so, we lower the bar for models to learn from demonstrations by adding more explicit information to each support data. We also consider a variety of datasets related to both general perception and specialized reasoning to make the study more comprehensive and trustworthy.

We focus on evaluating VLMs which has an open-source reasoning variant, such as Qwen2.5-VL (Bai et al., 2025) and VLM-R1 (Shen et al., 2025), VL-Rethinker (Wang et al., 2025a); InternVL2.5 (Chen et al., 2024c) and InternVL2.5-MPO (Wang et al., 2024); Llama-3.2V (MetaAI, 2024) and LLaVA-CoT (Xu et al., 2024a). We list the parameter size of each version in Appendix Tab. 5. Through

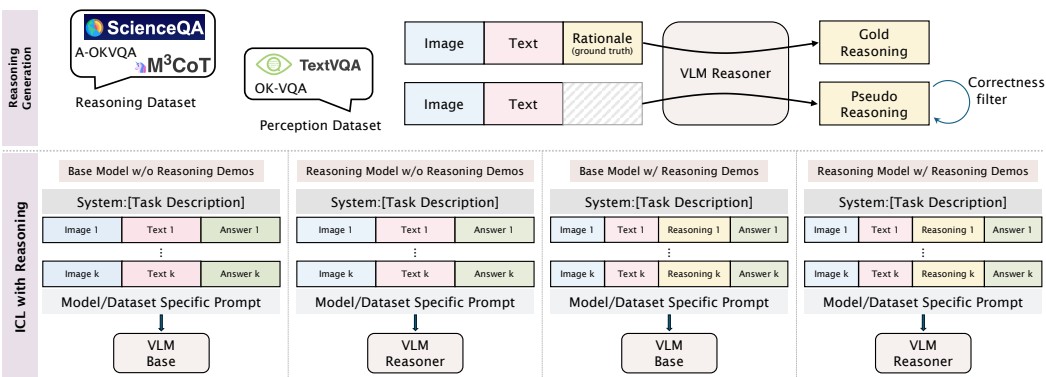

Figure 4: Visualization of Full Pipeline for ICL with VLM Reasoner

enforcing such a pairing relationship, we can better compare the ICL capabilities across models from a fair perspective. To evaluate the effects of reasoning components on MM-ICL performance, we consider four different protocols. The prompts for each protocol are depicted in Appendix in Fig. 8.

**(1) Base model without reasoning demos.** This is the standard setting of MM-ICL, where each demo is a concatenation of image, question, and ground truth answer.

**(2) Reasoning model without reasoning demos.** This setting is similar to Protocol (1), and serves as the standard MM-ICL baseline for reasoning models. As before, the support sample is a concatenation of image, question, and ground truth answer. The key difference is that the reasoning model is expected to generate both the final answer and a rationale.

**(3) Base model with reasoning demos.** This setting evaluates whether providing additional information (i.e., ground truth or generated rationales) in the support set helps the model learn from them and predict the correct answer for the query through ICL. If the dataset includes ground truth rationales, they are concatenated with the image and question in the support samples.

**(4) Reasoning model with reasoning demos.** This setting is similar to Protocol (3), except that the reasoning in the demo needs to be formatted in the same way as the reasoner model is trained. This is to address the issue of **format inconsistency**, which is discussed below.

We argue that the standard MM-ICL Protocol (2) for reasoning models introduces a **format inconsistency**. While reasoning models are trained with interleaved inputs—image, question, rationale, and answer—the demonstrations in MM-ICL typically include only the answer. In contrast, the model is expected to generate both the explanation and the answer for the query. This mismatch can lead to suboptimal performance, e.g., the model may focus solely on developing the answer and ignore the rationale, resulting in degraded output quality (similar findings in Zheng et al. (2025)) (see Sec. 5.1).

To address this, we newly introduce **Protocol (4)** for the best practice of MM-ICL with VLM reasoners, which contains a two-stage process. First, we prompt the model with each support sample to generate both a rationale (**Pseudo Reasoning**) and an answer, ensuring consistency with the expected output format. We then concatenate these generated reasoning-augmented demonstrations with the original input of each support to form a coherent and format-aligned context for the query.

However, since the generated rationales may vary in quality, we introduce two strategies to improve reliability: (1) Ground truth rationale reformulation: If ground truth rationales are available, we use them as input to the model to reformat the rationale into the desired structure (**Gold Reasoning**). (2) Correctness-based filtering: We use the correctness of the generated answer as a heuristic to filter out support samples with misleading rationales. The whole pipeline is illustrated in Fig. 4.

## 5 EXPERIMENTAL RESULTS

We consider datasets that focus on perception and reasoning in the following experiments.
**Perception Datasets.** TextVQA (Singh et al., 2019) and OK-VQA (Reichman et al., 2023) focus on reading text in images and answering commonsense questions, respectively. They use their own answer matching metrics (e.g., string normalization and consensus-based accuracy) for evaluation.

Table 1: Comparison of inconsistent and consistent support-query format with VLM reasoners.

| Ablation | A-OKVQA | | | | ScienceQA | | | | M$^3$CoT | | | |
|---|---|---|---|---|---|---|---|---|---|---|---|---|
| | 1 | 2 | 4 | 8 | 1 | 2 | 4 | 8 | 1 | 2 | 4 | 8 |
| **VLM-R1** | | | | | | | | | | | | |
| inconsistent | 81.31 | 81.31 | 80.44 | 79.39 | 81.71 | 81.95 | 81.31 | 80.61 | 51.64 | 53.36 | 51.51 | 50.60 |
| consistent | 82.45 | 81.83 | 81.48 | 81.14 | 82.30 | 83.39 | 82.45 | 82.94 | 53.19 | 53.80 | 54.36 | 52.89 |
| Δ | +1.14 | +0.52 | +1.04 | +1.75 | +0.59 | +1.44 | +1.14 | +2.33 | +1.55 | +0.44 | +2.85 | +2.29 |
| **VL-Rethinker-7B** | | | | | | | | | | | | |
| inconsistent | 85.76 | 85.24 | 84.28 | 83.76 | 89.04 | 88.94 | 88.89 | 89.14 | 66.22 | 67.60 | 66.39 | 66.82 |
| consistent | 85.50 | 84.98 | 84.98 | 85.68 | 90.23 | 90.23 | 90.18 | 90.23 | 68.08 | 69.15 | 68.59 | 68.55 |
| Δ | -0.26 | -0.26 | +0.70 | +1.92 | +1.19 | +1.29 | +1.29 | +1.09 | +1.86 | +1.55 | +2.20 | +1.73 |
| **LLaVA-CoT** | | | | | | | | | | | | |
| inconsistent | 85.85 | 85.33 | 84.28 | 83.14 | 91.52 | 90.98 | 87.65 | 84.78 | 55.26 | 51.42 | 44.26 | 42.28 |
| consistent | 86.20 | 85.59 | 84.54 | 83.23 | 92.81 | 91.97 | 91.57 | 90.48 | 54.62 | 53.62 | 52.29 | 50.99 |
| Δ | +0.35 | +0.26 | +0.26 | +0.09 | +1.29 | +0.99 | +3.92 | +5.70 | -0.64 | +2.20 | +8.03 | +8.71 |

**Reasoning Datasets.** ScienceQA (Lu et al., 2022), A-OKVQA (Schwenk et al., 2022), and M$^3$CoT (Chen et al., 2024a) target multi-step reasoning. ScienceQA features science questions accompanied by images and provides expert-written explanations as rationales. A-OKVQA offers curated natural language rationales aligned with commonsense reasoning. M$^3$CoT uses chain-of-thought rationales generated via prompting to guide multi-hop reasoning. We follow the VLMEvalKit (Duan et al., 2024) setup, which uses GPT-4o mini as a judge to assess answer quality.

For each dataset, we use the training split to construct the support set. The query set is taken from the test split if ground-truth answers are available; otherwise, we use the validation split. Unless otherwise specified, we use Protocol (1) for VLM base models and Protocol (4) for VLM reasoners.

## 5.1 FORMAT ALWAYS MATTERS: A CASE STUDY ON THE FORMAT INCONSISTENCY ISSUE

To assess whether modern VLMs still suffer from the mismatch in format of the MM-ICL demonstrations, we experiment with a reasoning-aware support-query format. Specifically, we compare two setups: **(1) Inconsistent** (Protocol (2) in Sec. 4): Each demonstration contains only the final answer. **(2) Consistent** (Protocol (4) in Sec. 4): Each demonstration contains the full reasoning process, including rationale and answer, mirroring the model's expected generation format.

To ensure a fair comparison, we use **Pseudo Reasoning**, where the rationale component in each demonstration is generated by the model itself based on the original support set inputs. This avoids the need for additional supervision and keeps all settings grounded in the same available information.

Tab. 1 shows results across three reasoning models (VLM-R1, VL-Rethinker-7B, and LLaVA-CoT) on three benchmarks (A-OKVQA, ScienceQA, M$^3$CoT) under different shots. Consistent formatting where demonstrations include both rationale and answer consistently outperforms inconsistent formatting across models and datasets, especially in high-shot settings (e.g., +8.71 on M$^3$CoT with 8 shots using LLaVA-CoT). This suggests that aligning the demonstration format with the model's expected output is crucial for effective MM-ICL with reasoning, even for capable model VLMs. **We stick to this consistent formatting pipeline for reasoning models for other experiments.**

## 5.2 DOES MM-ICL WITH REASONING HELP? ZERO-SHOT VS. FEW-SHOT

With the best practice for MM-ICL with VLM reasoners established, we now proceed to determine whether VLMs can successfully perform MM-ICL with information-enriched demos. We present the results in Tab. 2 and Tab. 3, and more in Appendix (full results in Tab. 11, Tab. 12 and results on ICL-tuned models in Tab. 10). As evident from the tables, in the majority of cases, MM-ICL with a few demonstrations does not exceed the performance when no demonstrations are presented. An improvement is observed in some rare cases, while the performance gain is often minimal. These results suggest that current VLMs/VLM reasoners indeed **can barely learn from demonstration data** even if the presented example data is ID. This demonstrates an essential weakness of current VLMs compared to their language-only counterparts, where ICL is often considered to be widely capable and beneficial (Baldassini et al., 2024; Qin et al., 2024).

The failure of VLMs in MM-ICL could stem from two factors: 1) the models lack the ability to learn from demonstrations, 2) the support rationales themselves are of low quality and therefore

Table 3: Reasoning datasets accuracy across models and shots. Higher accuracy between 0-shot and best few(1,2,4,8)-shot performance within the same model is **bolded**.

| Models | A-OKVQA | | ScienceQA | | M³CoT | |
|---|---|---|---|---|---|---|
| | 0-shot | best few-shot | 0-shot | best few-shot | 0-shot | best few-shot |
| Qwen2.5-VL-3B-Instruct | **85.41** | 82.01 | **81.61** | 81.11 | **51.77** | 51.34 |
| VLM-R1 | **85.07** | 82.45 | 82.30 | **83.39** | 53.11 | **54.36** |
| Qwen2.5-VL-7B-Instruct | 88.56 | **88.65** | **88.99** | 87.65 | **63.03** | 60.53 |
| VL-Rethinker-7B | **85.68** | **85.68** | 89.64 | **90.23** | 67.90 | **69.15** |
| Qwen2.5-VL-72B-Instruct | **91.44** | 91.18 | 91.18 | **91.57** | **70.23** | 70.02 |
| VL-Rethinker-72B | 88.82 | **89.34** | **94.40** | 93.75 | 74.85 | **76.40** |
| Llama-3.2-11B-Vision-Instruct | **84.02** | **84.02** | 83.99 | **84.43** | 42.45 | **43.74** |
| LLaVA-CoT | **87.42** | 86.20 | **94.55** | 92.81 | **56.26** | 54.62 |
| InternVL2.5-4B | **85.85** | 84.37 | **97.17** | 96.43 | **55.74** | 54.44 |
| InternVL2.5-4B-MPO | **84.89** | 83.58 | 97.32 | 96.88 | **64.50** | 58.54 |
| InternVL2.5-8B | **87.42** | 86.90 | **98.07** | 97.77 | **62.42** | 59.92 |
| InternVL2.5-8B-MPO | **87.25** | 86.03 | **98.56** | 98.22 | **73.51** | 68.98 |
| Gemini 2.0 Flash-non-thinking | 89.52 | **90.04** | 88.00 | **89.69** | 62.51 | **64.28** |
| Gemini 2.0 Flash-thinking | **91.27** | 90.66 | 91.47 | **92.46** | 71.40 | **74.68** |
| Gemini 2.5 Flash-non-thinking | 90.04 | **90.22** | 93.85 | 74.52 | **74.59** | 55.31 |
| Gemini 2.5 Flash-thinking | 90.39 | **90.48** | 95.14 | **95.19** | 72.48 | 72.00 |

uninformative. To reduce the possibility of 2), we enhance the rational capability by filtering incorrect samples and injecting ground-truth rationales, as shown in Tab. 4 (full tables in the Appendix).

We found that improving rationale quality, either by filtering out incorrect support samples or injecting ground truth rationale, does not consistently lead to improved performance. In several cases, applying filters to remove incorrect samples slightly degrades performance. One possible reason behind the performance drop is that the filtering operation causes a reduction in support sample diversity and coverage, suggesting that support set sufficiency and diversity may play a more critical role than the information quality for the current VLMs (Zhang et al., 2022; Qin et al., 2024). This implies that the evaluated VLMs are insensitive to the information quality in the demos, further reinforcing the conclusion that **VLMs still lack true MM-ICL capabilities to effectively learning from demonstration data**.

Table 2: Perception datasets accuracy across models and shots. Best values across shots in **bold**.

| Models | TextVQA | | OK-VQA | |
|---|---|---|---|---|
| | 0-shot | best few-shot | 0-shot | best few-shot |
| Qwen2.5-VL-3B-Instruct | **79.13** | 78.70 | 54.09 | **56.63** |
| VLM-R1 | **74.87** | 74.54 | 40.00 | **42.97** |
| Qwen2.5-VL-7B-Instruct | **85.39** | 84.79 | 58.74 | **62.68** |
| VL-Rethinker-7B | **76.46** | 73.01 | **32.43** | 30.14 |
| Llama-3.2-11B-Vision-Instruct | 53.87 | **74.63** | 20.05 | **44.03** |
| LLaVA-CoT | 72.85 | **74.39** | **48.91** | 47.46 |
| InternVL2.5-4B | 78.68 | **78.88** | 49.88 | **54.79** |
| InternVL2.5-4B-MPO | 72.85 | **72.90** | 42.12 | **42.55** |
| InternVL2.5-8B | **79.03** | 78.73 | 57.20 | **59.62** |
| InternVL2.5-8B-MPO | 73.36 | **73.67** | 44.49 | **49.01** |
| Gemini 2.0 Flash-non-thinking | 77.13 | **78.53** | 40.47 | **50.49** |
| Gemini 2.0 Flash-thinking | **77.87** | 76.64 | 41.17 | **43.63** |

Table 4: Comparison of the Quality of the Rationales. Filter: filter out the incorrect support samples. gt R: add ground truth rationale as the input for support set reasoning generation.

| Ablation | **A-OKVQA** | | | | **ScienceQA** | | | | **M³CoT** | | | |
|---|---|---|---|---|---|---|---|---|---|---|---|---|
| | 1 | 2 | 4 | 8 | 1 | 2 | 4 | 8 | 1 | 2 | 4 | 8 |
| **VL-Rethinker-7B** | | | | | | | | | | | | |
| baseline | 85.50 | 84.98 | 84.98 | 85.68 | 90.23 | 90.23 | 90.18 | 90.23 | 68.08 | 69.15 | 68.59 | 68.55 |
| +filter | -0.35 | +0.17 | +0.43 | -0.61 | -0.49 | -0.05 | +0.00 | -0.49 | -0.18 | -0.17 | +0.52 | -0.65 |
| +gt R | +0.35 | +0.43 | +0.17 | +0.35 | -0.10 | -0.44 | +0.30 | +0.15 | -0.95 | -1.42 | -0.26 | +0.00 |
| +gt R & filter | -0.35 | +1.05 | +0.96 | -0.09 | -0.49 | -0.49 | +0.20 | +0.35 | +0.04 | -0.82 | +1.38 | -0.99 |
| **InternVL2_5-8B-MPO** | | | | | | | | | | | | |
| baseline | 86.03 | 84.89 | 84.54 | 81.92 | 98.07 | 98.22 | 97.57 | 96.48 | 68.98 | 67.56 | 65.96 | 63.37 |
| +filter | +1.13 | +0.61 | +0.26 | +1.84 | -0.10 | -0.10 | -0.10 | -0.10 | -0.04 | +0.69 | -0.21 | -2.41 |
| +gt R | +0.78 | +1.05 | +0.09 | +1.92 | -0.05 | -0.10 | -0.25 | +0.35 | +0.99 | +0.00 | +1.43 | +0.39 |
| +gt R & filter | +0.52 | +0.52 | +0.61 | +2.18 | +0.00 | -0.65 | -0.25 | -0.05 | -0.04 | +0.56 | +0.48 | +0.00 |

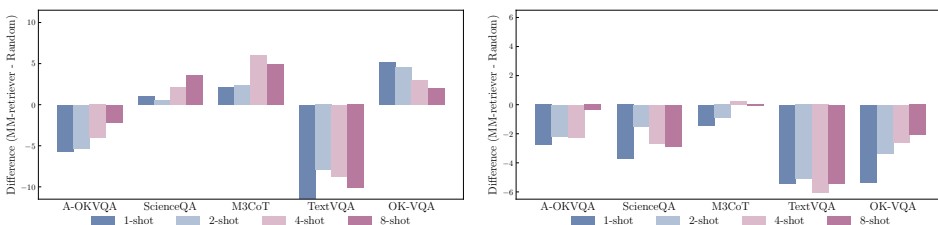

Figure 5: Comparison of Multimodal Retriever vs. Random Selection on 6 vision-language datasets. **Left:** Llama-3.2-11B-Vision-Instruct, **Right:** LLaVA-CoT

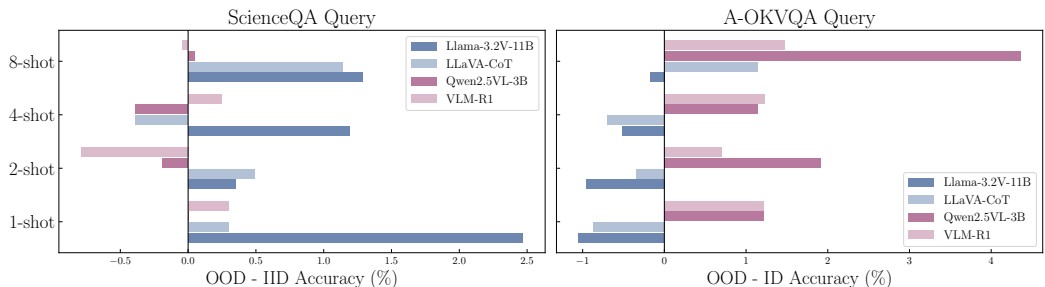

Figure 6: Performance difference between OOD and ID on ScienceQA and A-OKVQA

## 5.3 Assessing the Role of Retriever Methods

In Fig. 5, we compare the performance of the multimodal retriever (MM-retriever, details in App. A) against random selection. For base models, MM-retriever improves performance on M³CoT, ScienceQA and OK-VQA—suggesting that simple retrieval based on input similarity can be beneficial when no rationales are involved in the context. However, for reasoning models, MM-retriever consistently underperforms compared to random selection, especially on reasoning-intensive datasets.

Prior work has suggested that in-context learning often operates via majority voting or pattern matching over the demonstrations (Baldassini et al., 2024; Qin et al., 2024). In base models without explicit reasoning, retrieving demonstrations with similar inputs (e.g., similar image-question pairs) often yields support examples with highly similar answers. This facilitates a form of shallow copying, where the model infers the correct answer by identifying consistent patterns across demonstrations and can easily mimic the format or final answer from them, which outperforms random sampling, where the patterns of support and query samples are usually different.

However, for reasoning-augmented models, this heuristic breaks down. Even when the input similarity is high, the corresponding rationales can be diverse in content, structure, and logic chain. Because MM-retriever selects support examples based solely on input similarity (image and question), it does not account for whether the reasoning paths in the retrieved examples are consistent or relevant to the current query. As a result, the retrieved demonstrations may not form a coherent support set, making it harder for the model to extract useful patterns. In contrast, random sampling may introduce a more diverse set of rationales (Zhang et al., 2022), which while not tailored, can better expose the model to varied reasoning styles and prevent overfitting to a specific (and possibly irrelevant) reasoning trajectory. This may explain why MM-retriever underperforms random sampling in reasoning-intensive MM-ICL settings, despite its advantage in more shallow or pattern-based tasks.

## 5.4 ID versus OOD with Modern VLMs

To echo our original motivation and experiments in Sec. 3.2, we also benchmark the performances of MM-ICL with ID and OOD support sets on ScienceQA and A-OKVQA and summarize the results in Fig. 6. Unlike OpenFlamingo on TextVQA/OK-VQA, we notice a mixed trend across the number of shots between each model. We observe that for modern VLMs, when presented with the same number of demonstrations, it's possible that the OOD setting wins over the ID setting, regardless of datasets or models. This is possibly because, after we removed the format inconsistency, capable VLM/VLM reasoners are no longer easily misled by the answer format in the demos. With the use of LLM judges as answer evaluators, the conjectured performance gap between ID and OOD is further narrowed. However, we want to emphasize that these results do not imply the success of MM-ICL, since most of these few-shot results can't even match zero-shot performance. Interestingly, we also

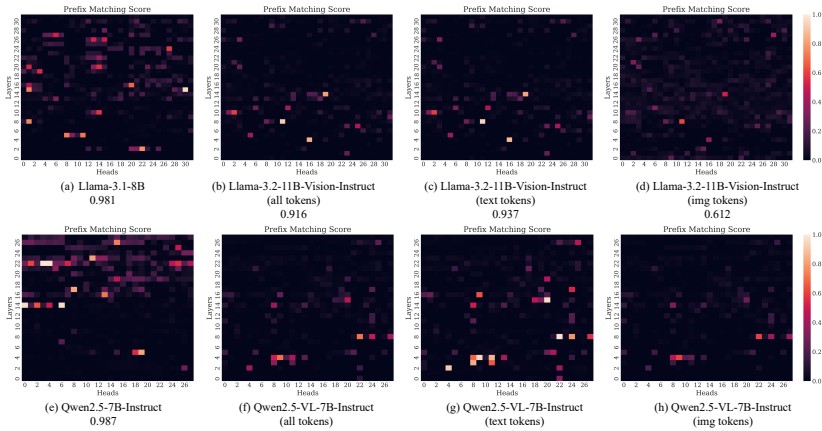

Figure 7: Prefix matching score heatmaps for LLaMA-3.1-8B-Instruct, LLaMA-3.2-11B-Vision-Instruct, Qwen2.5-7B-Instruct and Qwen2.5-VL-7B-Instruct.

notice that the winner of this duel between OOD and ID tends to be consistent among different shots, as well as among models of the same type. For example, on A-OKVQA, Qwen2.5-VL-3B consistently performs better with OOD demos, and its reasoner variant VLM-R1 also inherits this ability. A similar phenomenon is observed for the pair of Llama-3.2V-11B and LLaVA-CoT on both datasets. This further suggests that the MM-ICL capability of a VLM, including its robustness to OOD support data, is not significantly impacted during the RL training for reasoning.

# 6 WHY MM-ICL FALLS SHORT: AN ATTENTION-LEVEL PERSPECTIVE

To better understand why MM-ICL underperforms compared to its language-only counterpart, we examine the underlying attention mechanisms that enable ICL. Olsson et al. (2022) demonstrated that in LLMs, induction heads are specialized attention heads that appear to be the primary source of ICL. These heads operate by attending from the second occurrence of a token back to its earlier occurrence and then boosting the probability of the token that followed, allowing models to exploit repeated structures in context. To measure this effect for VLMs, we adopt the *prefix matching* protocol following Crosbie and Shutova (2025): we generate a sequence of 50 random tokens, excluding the most and least common tokens, repeat this sequence four times, and prepend a start-of-sequence token. We then compute the attention pattern and define the prefix matching score as the average attention mass from a given token back to the tokens that preceded it in earlier repeats. For VLMs, we adapt this setup by repeating the image and interleaving it with the text, and calculate the prefix matching scores for all image and text tokens, respectively. In Fig. 7, we found that LLMs exhibit a noticeable band of heads with high prefix scores. In contrast, its VLM counterparts exhibit lower prefix scores, indicating a weakening of the induction heads. Moreover, within VLMs, prefix matching is substantially stronger for text tokens (e.g., 0.937 for Llama-3.2-11B-V and 0.979 for Qwen2.5-VL-7B) than for image tokens (0.612 and 0.600), highlighting that image representations are especially deficient in supporting induction-like behavior. This gap potentially explains why MM-ICL falls short: VLMs struggle with prefix matching and therefore fail to leverage repeated multimodal demonstrations in the way LLMs do with text, thus cannot reliably exploit even basic patterns, let alone reasoning. Our results point to a deeper architectural limitation and suggest that extending induction-like mechanisms into multimodal attention may be crucial for robust MM-ICL.

# 7 CONCLUSION, LIMITATIONS AND FUTURE DIRECTION

Our study revisits the assumption that VLMs perform genuine MM-ICL. Under varying conditions and distribution shifts, we find that current VLMs often fail to utilize demonstrations meaningfully, relying instead on shallow cues. Our proposed MM-ICL with reasoning pipeline provides a stronger testbed; however, models exhibit limited sensitivity to shot count, retrieval method, and rationale quality. We further provide explanation by showing that weak prefix matching and absent induction heads underlie MM-ICL failure. **Limitations and future directions.** We focus on diagnosing MM-ICL weaknesses in VLMs without proposing architectural or training interventions. Future work could investigate architectural and scaling factors that enable induction-head-like mechanisms to emerge in multimodal settings, which may ultimately strengthen their ability to perform MM-ICL.

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

## A   MULTIMODAL IN-CONTEXT LEARNING

**Format of MM-ICL.** We denote the input quest $q$ from a query set $\mathcal{Q}$, with an image component $I_q$ and a question/instruction $T_q$. So we write $q = (I_q, T_q)$. For each query $q$, we define the associated context prompt $C_q$, consisting of $N$ demonstration examples from the support set $\mathcal{S}$ using an example selection protocol, defined as $C_q = \texttt{Retriever}(\mathcal{S}, N, q)$.

Each demonstration example includes image $I_i$, instruction $T_i$ and the response/answer $R_i$. For notation compactness, we write $C_q = \{(I_i, T_i, R_i)\}_{i=1,\ldots,N}$. We generate a response $R_q$ to each query $q$ using pre-trained VLMs as $R_q = \text{VLM}([p_{\text{sys}}, \texttt{Order}(C_q), p_{\text{instruct}}, q])$, where $p_{\text{sys}}$ is a system prompt that assigns personas or instructions so that the model generates responses in an intended format, $\texttt{Order}$ is a function that determines the sequential order of each demonstration example showing up in the model input, and $p_{\text{instruct}}$ is an additional, optional instruction prompt such as Chain-of-Thoughts (CoT) prompt (Wei et al., 2023). Researchers have found that the result of ICL is sensitive to the choice of $p_{\text{sys}}$, $\texttt{Order}$, and $p_{\text{instruct}}$, so we include them in the algorithmic framework of MM-ICL (Lu et al., 2021; Wu et al., 2022; Xiang et al., 2024; Qin et al., 2023).

**Design of `Retriever`.** A naive choice for the demonstration selection protocol is to choose the examples uniformly at random from the support set $\mathcal{S}$. While this avoids the risk of introducing bias into ICL, it also does not maximize the potential performance gain from this process by choosing specialized examples for different queries.

A common improvement over the random selection protocol is similarity-based retrievers, which score each data point in $\mathcal{S}$ by evaluating their similarity to the query and extract the most relevant ones. This procedure can be formally defined as the following: We start by defining a representation $h_i$ for each text-image pair $(I_i, T_i, R_i)$ as $h_i = \texttt{Encoder}(I_i, T_i, R_i)$. The common choice in the literature is often to use unimodal encoders to embed images $I_i$ and texts $T_i$ separately using models (Baldassini et al., 2024; Yang et al., 2022), and then combine them for final similarity computation. While, in principle, such a procedure incorporates information from both modalities, it overlooks the modality interaction that is crucial for selecting relevant examples, which potentially leads to suboptimal performance compared to using multimodal encoders (Xu et al., 2024b). Therefore, we utilize similarity-based retrievers with representations obtained from multimodal encoders to achieve a more competitive performance.

## B   PROMPT FORMAT FOR EACH MM-ICL PROTOCOL

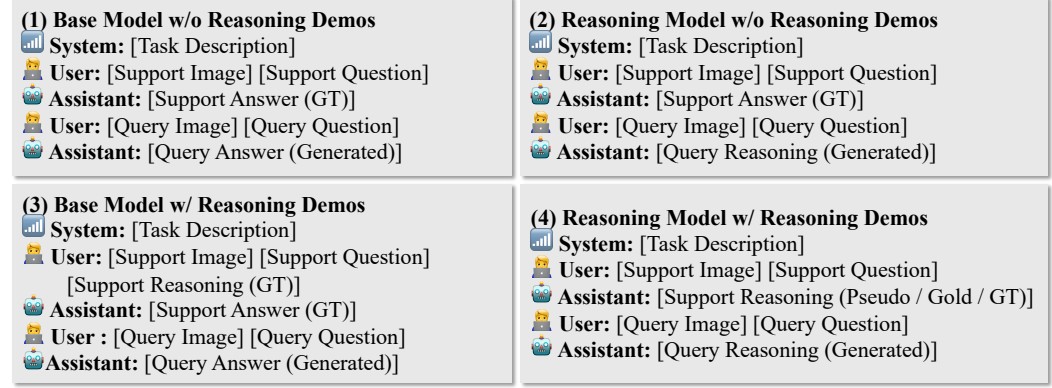

Figure 8: Prompt format for each MM-ICL protocol. GT stands for ground truth.

# C    MODEL SIZE

Table 5: VLMs and their associated reasoner version used in the evaluation.

| Base Model | Size | Reasoning Model |
|---|---|---|
| Qwen2.5-VL | 3B | VLM-R1 |
| | 7B | VL-Rethinker 7B |
| | 72B | VL-Rethinker 72B |
| InternVL 2.5 | 4B | InternVL 2.5-4B-MPO |
| | 8B | InternVL 2.5-8B-MPO |
| Llama-3.2V | 11B | LLaVA-CoT |

# D    ADDITIONAL RESULTS ON ID V.S. OOD FOR IDEFICS2 AND QWEN2.5-VL-3B-INSTRUCT

In Sec. 3.2, we present the results of MM-ICL for OpenFlamingo on TextVQA and OK-VQA for both In-Distribution (ID) and Out-of-Distribution (OOD) settings. Here, we include additional results for the same experiment setting, except for using IDEFICS2-8B and Qwen2.5-VL-3B-Instruct. The results for IDEFICS2-8B are presented in Tab. 6 and the results for Qwen2.5-VL-3B-Instruct are presented in Tab. 7.

Table 6: IDEFICS2-8B. Best values across shots are marked in **bold**.

| Query Dataset | Support Dataset | Method | 0-shot | 1-shot | 2-shot | 4-shot | 8-shot | 16-shot |
|---|---|---|---|---|---|---|---|---|
| TextVQA | TextVQA | Random | 64.62 | 66.87 | 67.00 | 67.77 | **68.37** | 67.54 |
| | | MM-Retriever | 64.62 | 64.38 | 63.83 | 64.18 | 65.86 | **65.94** |
| | OK-VQA | Random | 64.62 | 67.13 | 67.57 | 68.40 | 68.50 | **68.61** |
| | | MM-Retriever | 64.62 | 66.90 | 67.81 | 68.60 | **69.11** | 69.02 |
| OK-VQA | OK-VQA | Random | 56.48 | 56.72 | 56.81 | 57.36 | 57.75 | **57.99** |
| | | MM-Retriever | 56.48 | 55.75 | 56.08 | 55.66 | 57.65 | **59.52** |
| | TextVQA | Random | 56.48 | 56.02 | 55.88 | 56.17 | 56.23 | **56.51** |
| | | MM-Retriever | **56.48** | 55.95 | 55.60 | 56.26 | 56.12 | 56.22 |

Table 7: Qwen2.5-VL-3B-Instruct. Best values across shots are marked in **bold**.

| Query Dataset | Support Dataset | Method | 0-shot | 1-shot | 2-shot | 4-shot | 8-shot | 16-shot | 32-shot |
|---|---|---|---|---|---|---|---|---|---|
| TextVQA | TextVQA | Random | **79.13** | 78.70 | 78.58 | 78.09 | 77.80 | 78.06 | 77.92 |
| | | MM-Retriever | **79.13** | 76.49 | 76.44 | 76.56 | 76.42 | 76.23 | 76.40 |
| | OKVQA | Random | **79.13** | 78.58 | 77.92 | 77.91 | 77.66 | 77.38 | 76.85 |
| | | MM-Retriever | **79.13** | 76.20 | 75.84 | 75.73 | 75.63 | 75.47 | 75.08 |
| OK-VQA | OK-VQA | Random | 54.09 | **56.63** | 56.54 | 55.70 | 55.67 | 54.91 | 55.36 |
| | | MM-Retriever | 54.09 | 56.37 | 57.78 | 57.87 | 58.27 | 58.78 | **59.13** |
| | TextVQA | Random | 54.09 | **54.90** | 53.63 | 53.58 | 53.35 | 53.33 | 52.75 |
| | | MM-Retriever | **54.09** | 50.60 | 50.57 | 49.62 | 49.20 | 48.51 | 48.57 |

We see that more modern VLMs like Qwen2.5-VL barely benefit from MM-ICL as its zero-shot performance surpasses other settings in most cases. This suggests that capable VLMs can perform instruction-following well, often without the need for demonstration examples, and succeed in a zero-shot setting. However, it also indicates that VLMs fail to learn effectively from the demonstration examples, since the accuracy does not increase as the number of shots grows. For Qwen2.5-VL-3B-Instruct, we also observe that the performance difference between ID and OOD settings is quite minimal when using a random retriever for both datasets, which is well aligned with our hypothesis that the performance decrease in OOD for OpenFlamingo is merely due to model incapability and

answer format mismatch. Again, due to the strong answer formatting capability, Qwen2.5-VL-3B Instruct no longer suffers from the issue mentioned above, showing no significant performance difference between ID and OOD scenarios.

# E  ADDITIONAL RESULTS ON QUALITY OF RATIONALES

We have demonstrated in Sec. 5.2 that the quality of reasoning rationales does not have a significant impact on the evaluation results of VLM reasoners by presenting results on VL-Rethinker-7B and InternVL2.5-8B-MPO. Here, to further enhance the conclusion we draw and verify that it's universal across VLM reasoners, we present results on other VLM reasoners in Tab. 8. For other models, we also observe a similar phenomenon, where the correctness of reasoning doesn't play an important role in determining the model's performance through MM-ICL. This accords with our evaluation that modern VLM reasoners do not perform true MM-ICL by learning from the demonstration examples as the example quality seems to be an irrelevant factor in the evaluation.

To show that this is not a special problem for VLM reasoners only, but rather a general problem of modern VLMs, we perform additional evaluations to assess the effects of the quality of rationales for VLM base models. We present the results in Tab. 9. Here, for VLM base models, we adopt **protocol 3** when using ground truth rationales and **protocol 1** when not using them. For VLM reasoners, we always adopt **protocol 4**, with **Gold Reasoning** (ground truth rationale formatted in model's output format) when ground truth rationale is feasible, and **Pseudo Reasoning** when it's not. We see that regardless of whether ground truth rationales are incorporated, VLM base models do not enjoy performance gains from performing MM-ICL, given that zero-shot performances are mostly the best among all the presented accuracy numbers. This agrees with our findings on VLM reasoners, suggesting again that the failure of VLMs to perform true MM-ICL exists quite generally among modern VLMs.

Table 8: Comparison of the Quality of the Rationales. Filter: filter out the incorrect support samples. gt R: add ground truth rationale as the input for support set reasoning generation.

| Ablation | A-OKVQA | | | | ScienceQA | | | | M$^3$CoT | | | |
|---|---|---|---|---|---|---|---|---|---|---|---|---|
| | 1 | 2 | 4 | 8 | 1 | 2 | 4 | 8 | 1 | 2 | 4 | 8 |
| **VLM-R1** | | | | | | | | | | | | |
| baseline | 82.45 | 81.83 | 81.48 | 81.14 | 82.30 | 83.39 | 82.45 | 82.94 | 53.19 | 53.80 | 54.36 | 52.89 |
| +filter | +1.13 | +0.88 | -0.08 | -0.27 | -0.15 | -1.44 | -0.15 | -0.49 | -1.51 | +0.04 | -1.47 | +0.30 |
| +gt R | -0.44 | +0.35 | +0.62 | -0.00 | +0.60 | -0.35 | -0.10 | -0.29 | +0.43 | -0.00 | +0.56 | +1.29 |
| +gt R & filter | +1.31 | +1.23 | +0.79 | +0.43 | -0.05 | -0.30 | +0.59 | -0.89 | +0.43 | +0.77 | -0.95 | +1.94 |
| **VL-Rethinker-7B** | | | | | | | | | | | | |
| baseline | 85.50 | 84.98 | 84.98 | 85.68 | 90.23 | 90.23 | 90.18 | 90.23 | 68.08 | 69.15 | 68.59 | 68.55 |
| +filter | -0.35 | +0.17 | +0.43 | -0.61 | -0.49 | -0.05 | +0.00 | -0.49 | -0.18 | -0.17 | +0.52 | -0.65 |
| +gt R | +0.35 | +0.43 | +0.17 | +0.35 | -0.10 | -0.44 | +0.30 | +0.15 | -0.95 | -1.42 | -0.26 | +0.00 |
| +gt R & filter | -0.35 | +1.05 | +0.96 | -0.09 | -0.49 | -0.49 | +0.20 | +0.35 | +0.04 | -0.82 | +1.38 | -0.99 |
| **LLaVA-CoT** | | | | | | | | | | | | |
| baseline | 86.20 | 85.59 | 84.54 | 83.23 | 92.81 | 91.97 | 91.57 | 90.48 | 54.62 | 53.62 | 52.29 | 50.99 |
| +filter | -1.31 | -0.70 | +0.09 | +0.87 | -0.64 | +0.05 | +0.70 | +0.10 | -0.44 | +0.39 | +0.47 | +0.82 |
| +gt R | -0.79 | -0.87 | +0.44 | +0.18 | +0.70 | +0.15 | +0.05 | +0.84 | -0.74 | -0.77 | -0.05 | -0.21 |
| +gt R, filter | -1.31 | -0.87 | +1.49 | +1.22 | +0.50 | -0.35 | -0.25 | +0.55 | +1.07 | +1.21 | +1.38 | +1.51 |
| **InternVL2.5-4B-MPO** | | | | | | | | | | | | |
| baseline | 83.58 | 81.40 | 82.36 | 82.53 | 96.78 | 96.48 | 96.88 | 95.93 | 58.54 | 55.65 | 55.22 | 57.08 |
| +filter | +0.09 | +2.97 | +1.40 | +1.14 | -0.15 | +0.30 | -0.45 | +0.30 | +3.02 | +7.25 | +6.60 | +5.13 |
| +gt R | -0.26 | -0.70 | -2.19 | -2.09 | -0.35 | +0.40 | -0.55 | +0.05 | +2.85 | +3.84 | +1.51 | -1.77 |
| +gt R & filter | -0.09 | -2.01 | -2.27 | -3.67 | -0.15 | -0.10 | -0.30 | -0.05 | +2.59 | +3.67 | +2.03 | -1.60 |
| **InternVL2.5-8B-MPO** | | | | | | | | | | | | |
| baseline | 86.03 | 84.89 | 84.54 | 81.92 | 98.07 | 98.22 | 97.57 | 96.48 | 68.98 | 67.56 | 65.96 | 63.37 |
| +filter | +1.13 | +0.61 | +0.26 | +1.84 | -0.10 | -0.10 | -0.10 | -0.10 | -0.04 | +0.69 | -0.21 | -2.41 |
| +gt R | +0.78 | +1.05 | +0.09 | +1.92 | -0.05 | -0.10 | -0.25 | +0.35 | +0.99 | -0.00 | +1.43 | +0.39 |
| +gt R & filter | +0.52 | +0.52 | +0.61 | +2.18 | -0.00 | -0.65 | -0.25 | -0.05 | -0.04 | +0.56 | +0.48 | +0.00 |

Table 9: Effects of Ground Truth Rationale in MM-ICL of VLMs for Reasoning datasets. Best values across shots with or without ground truth rationales is in **bold**.

| Model | A-OKVQA | | | | | ScienceQA | | | | | M³CoT | | | | |
|---|---|---|---|---|---|---|---|---|---|---|---|---|---|---|---|
| | 0 | 1 | 2 | 4 | 8 | 0 | 1 | 2 | 4 | 8 | 0 | 1 | 2 | 4 | 8 |
| Qwen2.5-VL-3B-Instruct | **85.41** | 82.01 | 80.26 | 80.96 | 78.17 | **81.61** | 81.11 | 80.61 | 80.61 | 81.06 | 51.77 | 51.34 | 51.34 | 50.78 | 50.86 |
| + ground truth rationale | | 81.31 | 82.27 | 80.00 | 79.30 | | 81.11 | 80.86 | 81.11 | 81.06 | | 50.91 | 51.34 | **52.07** | 51.25 |
| VLM-R1 | **85.07** | 82.45 | 81.83 | 81.48 | 81.14 | 82.30 | 82.30 | **83.39** | 82.45 | 82.94 | 53.11 | 53.19 | 53.80 | 54.36 | 52.89 |
| + ground truth rationale | | 82.01 | 82.18 | 82.10 | 81.14 | | 82.90 | 83.04 | 82.35 | 82.65 | | 53.62 | 53.80 | **54.92** | 54.18 |
| Qwen2.5-VL-7B-Instruct | 88.56 | **88.65** | 87.86 | 87.77 | 88.03 | **88.99** | 86.71 | 87.65 | 86.56 | 86.86 | **63.03** | 60.40 | 60.05 | 60.22 | 60.53 |
| + ground truth rationale | | 88.12 | 88.38 | 87.60 | 87.51 | | 86.02 | 86.91 | 86.42 | 86.91 | | 61.09 | 60.70 | 61.35 | 60.87 |
| VL-Rethinker-7B | 85.68 | 85.50 | 84.98 | 84.98 | 85.68 | 89.64 | 90.23 | 90.23 | 90.18 | 90.23 | 67.90 | 68.08 | **69.15** | 68.59 | 68.55 |
| + ground truth rationale | | 85.85 | 85.41 | 85.15 | **86.03** | | 90.13 | 89.79 | **90.48** | 90.38 | | 67.13 | 67.73 | 68.33 | 68.55 |
| Llama-3.2-11B-Vision-Instruct | **84.02** | 84.02 | 83.49 | 83.58 | 82.71 | 83.99 | 82.85 | 84.43 | 84.13 | 83.64 | 42.45 | 42.75 | 43.74 | 42.58 | 42.49 |
| + ground truth rationale | | 83.32 | 83.84 | 83.23 | 83.67 | | **84.93** | 85.13 | 84.18 | 83.79 | | **44.43** | 42.97 | 42.58 | 42.15 |
| LLaVA-CoT | **87.42** | 86.20 | 85.59 | 84.54 | 83.23 | **94.55** | 92.81 | 91.97 | 91.57 | 90.48 | **56.26** | 54.62 | 53.62 | 52.29 | 50.99 |
| + ground truth rationale | | 85.41 | 84.72 | 84.98 | 83.41 | | 93.51 | 92.12 | 91.62 | 91.32 | | 53.88 | 52.85 | 52.24 | 50.78 |
| InternVL2.5-4B | **85.85** | 84.37 | 84.02 | 83.76 | 83.49 | **97.17** | 96.28 | 96.43 | 95.93 | 95.54 | **55.74** | 54.27 | 53.80 | 53.97 | 54.44 |
| + ground truth rationale | | 84.45 | 84.19 | 82.53 | 83.14 | | 96.53 | 96.68 | 96.53 | 95.49 | | 54.62 | 54.62 | 54.31 | 54.83 |
| InternVL2.5-4B-MPO | **84.89** | 83.58 | 81.40 | 82.36 | 82.53 | **97.32** | 96.78 | 96.48 | 96.88 | 95.93 | **64.50** | 58.54 | 55.65 | 55.22 | 57.08 |
| + ground truth rationale | | 83.32 | 80.70 | 80.17 | 80.44 | | 96.43 | 96.88 | 96.33 | 95.98 | | 61.39 | 59.49 | 56.73 | 55.31 |
| InternVL2.5-8B | **87.42** | 86.90 | 84.98 | 85.76 | 85.76 | **98.07** | 97.77 | 97.72 | 97.17 | 96.08 | **62.42** | 59.92 | 59.75 | 58.46 | 57.42 |
| + ground truth rationale | | 86.90 | 86.03 | 85.94 | 84.72 | | 97.82 | 97.57 | 97.32 | 96.43 | | 59.36 | 58.93 | 58.63 | 58.54 |
| InternVL2.5-8B-MPO | **87.25** | 86.03 | 84.89 | 84.54 | 81.92 | **98.56** | 98.07 | 98.22 | 97.57 | 96.48 | **73.51** | 68.98 | 67.56 | 65.96 | 63.37 |
| + ground truth rationale | | 86.81 | 85.94 | 84.63 | 83.84 | | 98.02 | 98.12 | 97.32 | 96.83 | | 69.97 | 67.56 | 67.39 | 63.76 |

## F ADDITIONAL RESULTS ON ICL-TUNED MODELS

In the main paper, we primarily focus on recent VLMs and VLM reasoners, as these models are expected to possess general in-context learning capabilities. In the Appendix, we also benchmark the performance of Otter (Li et al., 2023b;a) and MMICL (Zhao et al., 2023), which have explored Multimodal In-Context Instruction Tuning specifically to enhance ICL capabilities, and present the results in the table below. Similar to other models studied in this work, Otter and MMICL exhibit the same performance pattern, where they tend to underperform as the number of demonstrations increases. This suggests that the incapability of VLM to truly learning from the context potentially originates from some deeper, more fundamental aspects of VLM, and can't be simply fixed with instruction-tuning datasets.

Table 10: Results on A-OKVQA, ScienceQA, and M³CoT with ICL-Tuned Models.

| A-OKVQA | | | | | |
|---|---|---|---|---|---|
| Model | 0-shot | 1-shot | 2-shot | 4-shot | 8-shot | Best few-shot |
| Otter | **60.26** | 59.21 | 59.39 | 59.65 | 59.21 | 59.65 |
| Otter (w/ gt rationale) | 60.26 | 61.05 | 61.14 | 60.35 | 60.00 | **61.14** |
| MMICL | 76.24 | 69.34 | 75.28 | 76.42 | 24.10 | **76.42** |
| MMICL (w/ gt rationale) | **76.24** | 71.27 | 72.40 | 22.62 | 24.54 | 72.40 |
| **ScienceQA** | | | | | |
| Model | 0-shot | 1-shot | 2-shot | 4-shot | 8-shot | Best few-shot |
| Otter | **65.25** | 64.20 | 61.48 | 61.33 | 61.23 | 64.20 |
| Otter (w/ gt rationale) | **65.25** | 62.87 | 63.31 | 63.01 | 60.49 | 63.31 |
| MMICL | **76.30** | 71.29 | 69.16 | 43.13 | 30.74 | 71.29 |
| MMICL (w/ gt rationale) | **76.30** | 68.77 | 55.03 | 29.85 | 28.46 | 68.77 |
| **M³CoT** | | | | | |
| Model | 0-shot | 1-shot | 2-shot | 4-shot | 8-shot | Best few-shot |
| Otter | **35.12** | 34.38 | 33.35 | 32.01 | 33.95 | 34.38 |
| Otter (w/ gt rationale) | **35.12** | 34.86 | 33.78 | 32.21 | 31.17 | 34.86 |
| MMICL | **39.99** | 34.43 | 36.89 | 36.41 | 25.93 | 36.89 |
| MMICL (w/ gt rationale) | **39.99** | 32.53 | 26.27 | 26.27 | 25.75 | 32.53 |

## G RESULTS ON GENERAL AND REASONING DATASET

In this section, we present full results for the model used in evaluations on all five datasets, across different settings. The results on general datasets can be found in Tab. 11, and the results on reasoning datasets can be found in Tab. 12.

Note that we also include Gemini 2.0/2.5 Flash, closed-source, commercial-grade models. We use the Gemini 2.5 Flash native non-thinking and thinking components by setting the thinking budget to zero and non-zero. Both Gemini 2.0 Flash (non-thinking) and Gemini 2.0 Flash (thinking) use the same underlying model, but differ in prompting protocols—(1) and (4), respectively. For protocol (1), we use the prompt `"Answer the question directly."` For protocol (4), we use the prompt `"Give step-by-step reasoning before you answer, and when you're ready to answer, please use the format Final answer:  ..."` for both the Support Pseudo Reasoning generation/demonstrations and the query in MM-ICL. Overall, our conclusions also generalize well to proprietary models such as Gemini 2.0 and Gemini 2.5. As can be seen from the tables, as the number of demonstrations presented increases, the performance gain is minimal in most cases, which is consistent with our observations on the open-sourced VLM/VLM-reasoners. Similarly, we have also observed a performance degradation of Gemini when more demos are presented, highlighting again that modern VLM's failure of truly learning from the context is quite universal. For example, on M3CoT, the performance of Gemini 2.5 Flash drops from 74.59/72.48 to 55.31/72.00. These counterintuitive results agree with our findings that VLM lacks true ICL capabilities.

Table 11: Accuracy across models and shots for general datasets. Best values across shots are **bolded**. Best values across few-shots are underlined.

| Model | TextVQA | | | | | OK-VQA | | | | |
|---|---|---|---|---|---|---|---|---|---|---|
| | 0 | 1 | 2 | 4 | 8 | 0 | 1 | 2 | 4 | 8 |
| Qwen2.5-VL-3B-Instruct | **79.13** | _78.70_ | 78.58 | 78.09 | 77.80 | 54.09 | **_56.63_** | 56.54 | 55.70 | 55.67 |
| VLM-R1 | **74.87** | 74.29 | 74.33 | _74.54_ | 73.42 | 40.00 | 41.59 | **_42.97_** | 42.95 | 42.45 |
| Qwen2.5-VL-7B-Instruct | **85.39** | _84.79_ | 84.45 | 84.09 | 84.12 | 58.74 | 62.33 | **_62.68_** | 62.49 | 62.33 |
| VL-Rethinker-7B | **76.46** | _73.01_ | 72.38 | 72.04 | 72.32 | **32.43** | _30.14_ | 29.37 | 29.04 | 28.86 |
| Llama-3.2-11B-Vision-Instruct | 53.87 | 74.52 | **_74.63_** | 74.43 | 73.83 | 20.05 | 37.49 | 40.05 | 42.89 | **_44.03_** |
| LLaVA-CoT | 72.85 | **_74.39_** | 74.04 | 74.39 | 72.96 | **48.91** | _47.46_ | 46.95 | 46.33 | 45.61 |
| InternVL2.5-4B | 78.68 | **_78.88_** | 78.53 | 77.77 | 77.39 | 49.88 | 54.68 | **_54.79_** | 54.41 | 53.63 |
| InternVL2.5-4B-MPO | 72.85 | **_72.90_** | 72.78 | 71.64 | 71.51 | 42.12 | 41.85 | **_42.55_** | 41.54 | 42.05 |
| InternVL2.5-8B | **79.03** | _78.73_ | 78.61 | 78.45 | 77.65 | 57.20 | **_59.62_** | 59.13 | 58.30 | 57.51 |
| InternVL2.5-8B-MPO | 73.36 | **_73.67_** | 73.37 | 72.47 | 73.03 | 44.49 | 48.11 | **_49.01_** | 48.51 | 47.98 |
| Gemini 2.0 Flash-non-thinking | 77.13 | 72.09 | 74.86 | 77.48 | **_78.53_** | 40.47 | 45.12 | 47.54 | 49.49 | **_50.49_** |
| Gemini 2.0 Flash-thinking | **77.87** | 71.65 | 75.11 | 75.77 | _76.64_ | 41.17 | **_43.63_** | 42.83 | 43.14 | 42.64 |

Table 12: Accuracy across models and shots for reasoning datasets. Best values across shots are **bolded**. Best values across few-shots are underlined.

| Model | A-OKVQA | | | | | ScienceQA | | | | | M³CoT | | | | |
|---|---|---|---|---|---|---|---|---|---|---|---|---|---|---|---|
| | 0 | 1 | 2 | 4 | 8 | 0 | 1 | 2 | 4 | 8 | 0 | 1 | 2 | 4 | 8 |
| Qwen2.5-VL-3B-Instruct | 85.41 | _82.01_ | 80.26 | 80.96 | 78.17 | 81.61 | _81.11_ | 80.61 | 80.61 | 81.06 | 51.77 | _51.34_ | 51.34 | 50.78 | 50.86 |
| VLM-R1 | 85.07 | _82.45_ | 81.83 | 81.48 | 81.14 | 82.30 | 82.30 | **83.39** | 82.45 | 82.94 | 53.11 | 53.19 | **_53.80_** | 54.36 | 52.89 |
| Qwen2.5-VL-7B-Instruct | 88.56 | **_88.65_** | 87.86 | 87.77 | 88.03 | **88.99** | 86.71 | _87.65_ | 86.56 | 86.86 | 63.03 | **_60.40_** | 60.05 | 60.22 | 60.53 |
| VL-Rethinker-7B | 85.68 | 85.50 | 84.98 | 84.98 | _85.68_ | 89.64 | **90.23** | _90.23_ | 90.18 | 90.23 | 67.90 | 68.08 | **_69.15_** | 68.59 | 68.55 |
| Qwen2.5-VL-72B-Instruct | 91.44 | 91.18 | 90.83 | 91.18 | 90.39 | 91.18 | 91.18 | 91.47 | **_91.57_** | 91.57 | 70.23 | 68.94 | 68.90 | _70.02_ | 68.42 |
| VL-Rethinker-72B | 88.82 | **_89.34_** | 89.17 | 89.17 | 88.30 | 94.40 | _93.75_ | 93.06 | 93.16 | 93.41 | 74.85 | 76.19 | 76.36 | 76.32 | **_76.40_** |
| Llama-3.2-11B-Vision-Instruct | 84.02 | _84.02_ | 83.49 | 83.58 | 82.71 | 83.99 | 82.85 | **84.43** | 84.13 | 83.64 | 42.45 | 42.75 | **_43.74_** | 42.58 | 42.49 |
| LLaVA-CoT | 87.42 | _86.20_ | 85.59 | 84.54 | 83.23 | 94.55 | _92.81_ | 91.97 | 91.57 | 90.48 | 56.26 | _54.62_ | 53.62 | 52.29 | 50.99 |
| InternVL2.5-4B | 85.85 | _84.37_ | 84.02 | 83.76 | 83.49 | 96.28 | 96.43 | _96.43_ | 95.93 | 95.54 | 55.74 | 54.27 | 53.80 | 53.97 | _54.44_ |
| InternVL2.5-4B-MPO | 84.89 | _83.58_ | 81.40 | 82.36 | 82.53 | 97.32 | 96.78 | 96.48 | **_96.88_** | 95.93 | 64.50 | _58.54_ | 55.65 | 55.22 | 57.08 |
| InternVL2.5-8B | 87.42 | _86.90_ | 84.98 | 85.76 | 85.76 | 98.07 | _97.77_ | 97.72 | 97.17 | 96.08 | 62.42 | _59.92_ | 59.75 | 58.46 | 57.42 |
| InternVL2.5-8B-MPO | 87.25 | _86.03_ | 84.89 | 84.54 | 81.92 | 98.56 | _98.07_ | 98.22 | 97.57 | 96.48 | 73.51 | _68.98_ | 67.56 | 65.96 | 63.37 |
| Gemini 2.0 Flash-non-thinking | 89.52 | 89.08 | 89.52 | **_90.04_** | 89.78 | 88.00 | 88.30 | 88.80 | 88.75 | **_89.69_** | 62.51 | 62.08 | 64.19 | 64.11 | **_64.28_** |
| Gemini 2.0 Flash-thinking | **91.27** | 89.96 | 89.52 | _90.66_ | 90.39 | 91.47 | 92.07 | 92.17 | **_92.46_** | 92.46 | 71.40 | 73.64 | 73.94 | 73.77 | **_74.68_** |
| Gemini 2.5 Flash-non-thinking | 90.04 | 70.83 | 88.73 | **_90.22_** | 89.61 | **93.85** | 60.54 | _74.52_ | 73.08 | 74.27 | **74.59** | 51.86 | _55.31_ | 50.95 | 51.51 |
| Gemini 2.5 Flash-thinking | 90.39 | 89.78 | 90.22 | **_90.48_** | 90.04 | 95.14 | 93.06 | 94.55 | 95.09 | **_95.19_** | 72.48 | 70.92 | 71.83 | _72.00_ | 71.10 |

## H  QUALITATIVE EXAMPLES OF THE PROMPT AND REASONING

In this section, we present some examples from the evaluation dataset we use to better demonstrate in detail the protocols we used for model evaluation. The examples are in Fig. 9 and Fig. 10. Note

that for models with different reasoning formats, we generate reasoning in the same formats for the demonstration examples. For example, Gemini 2.0 Flash outputs reasoning in steps, but LLaVA-CoT wraps the thinking process between the tags `<SUMMARY>` & `</SUMMARY>`, `<CAPTION>` & `</CAPTION>`, `<REASONING>` & `</REASONING>`, and `<CONCLUSION>` & `</CONCLUSION>`.

# I  SUMMARY OF TRAINING DATASETS FOR VLMS

We summarize the situation of each evaluation dataset in the training of the VLMs in Tab. 13. Such information can help the reader better judge the performance of each model across datasets and compare accuracy across models more fairly.

Table 13: Training datasets. ✓* indicates the dataset went through specific data processing pipelines before they were used in the training/finetuning. − represents that no information is found.

| Model | Size | OKVQA | TextVQA | A-OKVQA | ScienceQA | M³CoT |
|---|---|---|---|---|---|---|
| Qwen2.5-VL | 3B | − | − | − | − | − |
| VLM-R1 | 3B | | | | | |
| VL-Rethinker-7B | 7B | ✓* | | ✓* | ✓* | ✓* |
| VL-Rethinker-72B | 72B | ✓* | | ✓* | ✓* | ✓* |
| InternVL2.5 | 4B | ✓ | ✓ | ✓ | ✓ | |
| InternVL2.5-4B-MPO | 4B | ✓* | ✓* | | ✓* | ✓* |
| InternVL2.5-8B-MPO | 8B | ✓* | ✓* | | ✓* | ✓* |
| Llama-3.2V | 11B | − | − | − | − | − |
| LLaVA-CoT | 11B | | | | ✓ | ✓ |

# J  FURTHER DISCUSSION ON THE POTENTIAL CAUSE OF MMICL FAILURE

## J.1  MODEL ARCHITECTURE

To aid the discussion, we provide a summary table of the evaluated models, including their Year, Architecture, Parameters, Vision Encoder, Glue Layer, LLM, and MoE (Mixture-of-Expert) configurations.

Table 14: Comparison of recent vision-language models.

| Model | Year | Architecture | Parameters | Vision Encoder | Glue Layer | LLM | MoE |
|---|---|---|---|---|---|---|---|
| OpenFlamingo | 2023 | Decoder-only | 3B | CLIP ViT-L/14 (300M) | Cross-Attn | MPT-1B | None |
| IDEFICS-2 | 2024 | Decoder-only | 8B | SigLIP-SO (400M) | 2-layer MLP | Mistral-7B | None |
| Qwen2.5-VL | 2025 | Decoder-only | 3B/7B/72B | Redesigned ViT (500M) | 2-layer MLP | Qwen2.5-3B/7B/72B | None |
| InternVL-2.5 | 2025 | Decoder-only | 4B/8B | InternViT (300M) | 2-layer MLP | Qwen2.5-3B InternLM-2.5-7B | None |
| Llama-3.2-Vision | 2024 | Decoder-only | 11B | ViT-H/14 (630M) | Cross-Attn | LLaMA-3.1-8B | None |
| Gemini 2.0 | 2024 | Decoder-only | Undisclosed | Undisclosed | Undisclosed | Undisclosed | Undisclosed |
| Gemini 2.5 | 2025 | Decoder-only | Undisclosed | Undisclosed | Undisclosed | Undisclosed | Sparse MoE |

From Tab. 14 in the paper and the additional results provided in the following section, we find that models such as Gemini 2.0, Gemini 2.5 (thinking mode), and LLaMA-3.2-Vision appear more resilient to the common MM-ICL failure mode where few-shot performance is worse than zero-shot.

One potential reason may be related to their use of *cross-attention glue layers* such as in LLaMA-3.2-Vision, which could allow better token-level fusion between modalities. In contrast, models using 2-layer MLP glue, like InternVL-2.5 and Qwen2.5-VL, tend to show more frequent performance drops in few-shot settings.

We also speculate that *vision encoder strength* may play a role. Models using larger and more optimized encoders—such as ViT-H/14 (630M) in LLaMA-3.2-Vision and Redesigned ViT (500M) in Qwen2.5-VL—tend to perform more robustly than those with smaller encoders like CLIP ViT-L/14 (300M) or InternViT (300M). These encoders may produce richer visual features that are easier to align with the language model.

Additionally, *while larger model scale sometimes helps, it does not appear sufficient on its own*. For instance, Qwen2.5-VL-72B shows strong 0-shot but minor gains in few-shot.

Meanwhile, *VLM reasoners—such as VLM-R1, VL-Rethinker (7B and 72B, and Gemini 2.0/2.5 (thinking mode)—tend to be more resilient in few-shot settings*. We speculate this may be due to their use of reinforcement learning-based policy optimization techniques (e.g., GRPO), which could help the models better leverage in-context examples and mitigate the typical few-shot degradation.

Finally, regarding *MoE*, Gemini 2.5 (which uses a sparse MoE) performs well when "thinking" is enabled, but shows noticeable degradation in non-thinking mode. This may suggest that MoE may not inherently improve MM-ICL performance unless paired with reasoning mechanisms that can take advantage of the increased model capacity.

In summary, while we cannot make definitive claims, these patterns suggest that *glue layer design, vision encoder strength, and reasoning strategy* may all contribute to stronger few-shot performance in MM-ICL tasks.

### J.2    COMPARISON WITH TEXT-ONLY LLMS

There is evidence that even in text-only settings, *LLMs do not always effectively utilize demonstrations*. As shown in Zhong et al. (2024) (Tables 2 and 3), few-shot prompting often provides only marginal improvement over zero-shot performance across a range of tasks. Reynolds and McDonell (2021) further argues that in tasks like translation, the model may not learn anything new from a small number of demonstrations, and is instead directing the model to pre-learned information.

*When reasoning traces (i.e., rationale or CoT) are added, the situation does not always improve*. In Kojima et al. (2022), Figure 8 shows that for many datasets, CoT prompting in the few-shot setting offers no additional benefit over zero-shot CoT. Min et al. (2022) (Table 5) shows that the performance gain becomes much less when using examples with different answer types than with similar ones, confirming prior work Wang et al. (2022) which suggests that LLMs mostly leverage the few-shot examples to infer the repeated format rather than the task itself in-context. Most surprisingly, Sprague et al. (2025) demonstrates that CoT reasoning can still emerge even when the demonstrations include invalid or incorrect reasoning steps-prompting with invalid reasoning steps can achieve over 80-90% of the performance obtained using CoT, while still generating coherent lines of reasoning during inference.

A possible explanation offered by Zhong et al. (2024) is that *instruction tuning and human alignment have already taught the model to match task formats, reducing the value of few-shot demonstrations*. Since ICL often relies on similar format matching, its additional benefit becomes limited—further supporting our observation that demonstrations may act more as cues than sources of new information. Sprague et al. (2025) further suggests that the model's CoT behavior is often pretrained, and demonstrations serve only as triggers for latent capabilities rather than learning signals.

However, we also observe that *the failure of ICL is more pronounced in multimodal settings*. This indicates that, beyond the inherent issues in LLMs, additional bottlenecks likely exist in the visual encoder and the vision-language fusion module. Misalignment in visual representations, or the inability of the fusion module to generalize from multimodal context, could further impair the model's ability to benefit from in-context examples. Additionally, some forms of visual reasoning cannot be adequately expressed through text-only CoT, suggesting that developing *multimodal CoT* approaches—where visual and textual reasoning are jointly modeled—may be a promising direction to better support in-context learning in VLMs.

In summary, the failure of MMICL appears to stem from both the *inherent limitations of LLMs in utilizing demonstrations* and additional challenges introduced by the *visual encoder and multimodal fusion*, suggesting that both aspects contribute to the observed degradation and warrant deeper investigation.

## K    THE USE OF LARGE LANGUAGE MODELS (LLMS)

We use Large Language Models (LLMs) to aid and polish the writing. They were not involved in research ideation, experiment design, implementation, or analysis.

## Gemini 2.0 Flash w/ Pseudo Reasoning Demos on M³CoT (2-shot & random selection)

**User:** [Support Sample 1]
**Image**:

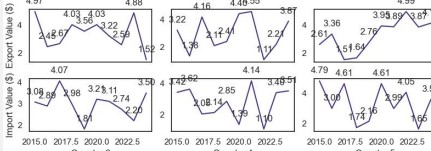

**Question**: In the provided image, how many items from the given choices will submerge underwater? Options: A. 0 B. 1 C. 2 D. 3 Please select the correct answer from the options above. Give step by step reasoning before you answer, and when you are ready to answer, please use the format "Final answer: .."

**Assistant:**
Here's how we can determine which objects will sink:
1. **Density Calculation:** An object will sink if its density is greater than the density of water (approximately 1000 kg/m³). Density is calculated as mass/volume.
2. **Calculate the density of each object:** * Object A: 5108.68 kg / 8.92 m³ = 572.72 kg/m³ * Object B: 5521.41 kg / 14.11 m³ = 391.31 kg/m³ * Object C: 4973.41 kg / 10.2 m³ = 487.59 kg/m³ * Object D: 2472.27 kg / 16.97 m³ = 145.68 kg/m³ * Object E: 7085.55 kg / 19.08 m³ = 371.36 kg/m³ * Object F: 2128.44 kg / 16.45 m³ = 129.39 kg/m³ * Object G: 3982.11 kg / 10.24 m³ = 388.88 kg/m³
3. **Compare to Water Density:** None of the objects have a density greater than 1000 kg/m³.
4. **Conclusion:** Since none of the objects have a density greater than water, none of them will sink.
Final answer: A

**User:** [Support Sample 2]
**Image**:

**Question**: How many countries have achieved fiscal surpluses in 2020? Options: A. 2 B. 5 C. 6 D. 7 E. 4 Please select the correct answer from the options above. Give step by step reasoning before you answer, and when you are ready to answer, please use the format "Final answer: .."

**Assistant:**
Here's how we can determine the number of countries with fiscal surpluses in 2020:
1. **Understanding Fiscal Surplus:** A fiscal surplus occurs when export value exceeds import value. In the graphs, we need to compare the export value (top graph) with the import value (bottom graph) for each country in the year 2020.
2. **Analyzing Each Country:** * **Country 3:** Export value in 2020 is approximately 4.03, and import value is approximately 3.23. Export > Import, so Country 3 has a surplus. * **Country 4:** Export value in 2020 is approximately 2.41, and import value is approximately 2.85. Export < Import, so Country 4 does not have a surplus. * **Country 5:** Export value in 2020 is approximately 3.89, and import value is approximately 2.99. Export > Import, so Country 5 has a surplus.
3. **Counting Surpluses:** Countries 3 and 5 have fiscal surpluses in 2020.
4. **Total:** Therefore, 2 countries have fiscal surpluses in 2020.
Final answer: A

**User:**[Query]
**Image**:

Suppose that $a^2$ varies inversely with $b^3$. If $a = 7$ when $b = 3$, find the value of $a^2$ when $b = 6$.

Choices:

(A) 5

(B) 7

(C) 0

(D) 6.125

**Question**: Based on the information in the picture, can you determine the correct solution for the Algebra problem? Options: A. option D in the image is the answer. B. the correct choice is option A in the image. C. the answer is option C in the image D. the correct choice is option B in the image. Please select the correct answer from the options above. Give step-by-step reasoning before you answer, and when you are ready to answer, please use the format "Final answer: .."

**Assistant:**
Here's how to solve the problem and determine the correct answer:
1. **Inverse Variation:** The statement "a² varies inversely with b³" means that a² * b³ = k, where k is a constant.
2. **Find the Constant (k):** We are given that a = 7 when b = 3. Therefore, a² = 7² = 49. So, k = a² * b³ = 49 * 3³ = 49 * 27 = 1323.
3. **Find a² when b = 6:** We know that a² * b³ = k, so a² = k / b³. When b = 6, a² = 1323 / 6³ = 1323 / 216 = 6.125.
4. **Match with the Choices:** The value of a² when b = 6 is 6.125, which corresponds to option D in the image.
Final answer: A

Figure 9: Gemini 2.0 Flash w/ Pseudo Reasoning Demos on M³CoT (2-shot & random selection)

## LLaVA-CoT w/ Pseudo Reasoning Demos on ScienceQA (2-shot & random selection)

**User:** [Support Sample 1]
**Image:**

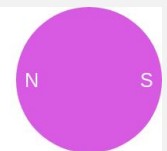

**Question**: Which country is highlighted? Options: A. Cuba B. Saint Vincent and the Grenadines C. Grenada D. Trinidad and Tobago Please select the correct answer from the options above.

**Assistant:**
<SUMMARY> I will analyze the map to identify the highlighted country and compare it with the options provided to determine the correct answer. </SUMMARY>
<CAPTION> The image depicts a map of the Caribbean region with several islands. One island, located at the southernmost part of the Caribbean, is highlighted in green. </CAPTION>
<REASONING> To identify the highlighted country, I will examine its location in the Caribbean. The highlighted area is at the southernmost part of the Caribbean, near the northeastern coast of South America. This location corresponds to Trinidad and Tobago, which is situated at the southern end of the Caribbean island chain. </REASONING>
<CONCLUSION> D </CONCLUSION>

**User:** [Support Sample 2]
**Image:**

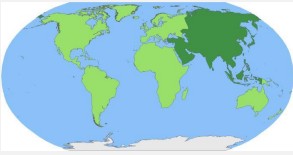

**Question**: Two magnets are placed as shown. Will these magnets attract or repel each other? Options: A. repel B. attract. Please select the correct answer from the options above.

**Assistant:**
<SUMMARY> To solve the problem, I will analyze the orientation of the magnets in the image and determine whether they will attract or repel each other based on the positions of their poles. </SUMMARY>
<CAPTION> The image shows two identical circular magnets. Each magnet has a distinct north pole (N) and south pole (S). The north pole of the left magnet is facing the south pole of the right magnet. </CAPTION>
<REASONING> Magnets have two poles: north (N) and south (S). Opposite poles attract each other, while like poles repel each other. In the image, the north pole of the left magnet is adjacent to the south pole of the right magnet. Since opposite poles (N and S) are facing each other, the magnets will attract. </REASONING>
<CONCLUSION> B. attract </CONCLUSION>

**User:** [Query]
**Image:**

**Question**: Which continent is highlighted? Options: A. Africa B. North America C. South America D. Asia. Please select the correct answer from the options above.

**Assistant:**
<SUMMARY> I will examine the image to identify the highlighted continent. I will describe the relevant details in the CAPTION, use logical reasoning to determine the continent in the REASONING section, and provide the final answer in the CONCLUSION. </SUMMARY>
<CAPTION> The image is a world map where one continent is highlighted in a darker shade. The highlighted continent is located in the eastern hemisphere and is known for its vast landmass. </CAPTION>
<REASONING> To identify the highlighted continent, I will consider its geographical position and characteristics. The highlighted area is in the eastern hemisphere, extending from the Middle East to East Asia. It is the largest landmass in this hemisphere, covering a vast area that includes regions such as Russia, China, and India. This description matches the continent of Asia. </REASONING>
<CONCLUSION> D. Asia </CONCLUSION>

Figure 10: LLaVA-CoT w/ Pseudo Reasoning Demos on ScienceQA (2-shot & random selection)

