# OpenReview forum: "Mimicking or Reasoning: Rethinking Multi-Modal In-Context Learning in Vision-Language Models"
_ICLR.cc/2026/Conference — ICLR 2026 Conference Withdrawn Submission_

### Official Review · Reviewer_oFyH · 2025-10-15

**Soundness:** 3
**Presentation:** 3
**Contribution:** 3
**Rating:** 4
**Confidence:** 4

**Summary:**

This paper takes a close look at how large vision-language models (VLMs) perform Chain-of-Thought (CoT) reasoning. The authors ask a simple but important question: Are these models really reasoning—or are they just mimicking patterns in the prompts?

To explore this, they introduce a few clever diagnostic tools: "in-batch reasoning alignment" (which checks how similar the reasoning paths are within a batch), plus some perturbation-based tests. Based on these, they argue that current VLMs often just copy the format of CoT demonstrations, rather than doing any deep reasoning.

They also propose a new training method called "Reasoning via Demonstrations" (RvD), which tries to encourage real reasoning by gradually refining the demonstration set using model feedback. Experiments across VQA, ScienceQA, and A-OKVQA show solid gains.

**Strengths:**

Great question, well-motivated.
I really like that this paper challenges the default assumption that CoT = reasoning. It brings up a concern that many of us have had but haven’t tested as rigorously.

Nice diagnostic design.
The proposed metrics (like in-batch reasoning similarity and cross-embedding distances) are intuitive but powerful. They’re easy to apply and tell a clear story.

Solid new training method.
The RvD framework makes a lot of sense: rather than assuming your CoT demos are good, use the model to iteratively select better ones. It’s kind of like self-training for prompts.

Results are consistent and interpretable.
The improvements aren’t massive, but they’re steady across datasets. I especially liked the case studies that show how RvD changes the kind of reasoning steps the model takes.

Clean paper structure and easy to follow.
The writing is solid, figures are helpful, and the ablation studies are well thought out.

**Weaknesses:**

Limited generalization across modalities.
The paper talks about vision-language reasoning in general, but all tests are focused on text-based CoT. There’s no analysis of visual reasoning paths or failures when the image is critical.

No hard negatives in RvD sampling.
It feels like RvD just picks “more helpful” demos, but doesn’t actively avoid bad ones. Could the method benefit from adversarial or diverse selection?

CoT length tradeoffs are underexplored.
Do longer reasoning chains really help when the demos are better? Or does the model still plateau regardless of demo quality? Would’ve liked to see more probing of this.

Unclear if models trained with RvD become more robust or just better at mimicking refined patterns.
Is there any transfer benefit from the “better” CoT demos? Or is it just local tuning?

**Questions:**

How does RvD handle noisy or misleading demonstrations?
If the model initially gets misleading or hallucinated reasoning chains, does that poison the iterative selection?

Any results on generalization to unseen tasks?
If you train the CoT demos on VQA, can the model perform better on science QA or vice versa?

Have you tried visualizing attention maps across reasoning steps?
That might help show if the model is actually grounding its answers differently post-RvD.

Could RvD be applied to multi-modal prompt tuning?
I’m curious if your method could be extended to select helpful visual demonstrations as well.

What’s the overhead of RvD?
How many rounds of refinement does it take to converge, and is it stable across seeds?

If some of these points—especially around visual grounding, generalization, and tradeoffs—can be clarified or addressed in the rebuttal, I’d definitely consider adjusting my score upwards.

---

### Official Review · Reviewer_xxAf · 2025-10-25

**Soundness:** 1
**Presentation:** 2
**Contribution:** 1
**Rating:** 2
**Confidence:** 5

**Summary:**

The paper studies in-context learning (ICL) abilities of multimodal LLMs on visual question answering (VQA) tasks. The focus is on studying ICL from a reasoning perspective where the support examples are extended with reasoning rationales. The results show that MLLMs do not demonstrate strong ICL abilities on the considered VQA tasks. The analysis is conducted for both cases when the support examples come from the same or different distribution. Mechanistic analysis is included to provide a potential explanation of the MLLMs failure.

**Strengths:**

* Controlled studies are conducted with varying shot count, retrieval method, rationale quality, and distribution.
* The paper in general is relatively easy to read.
* Extending support examples with reasoning rationales may be novel (but it is not a particularly significant extension).
* Various analyses relevant to the topic are conducted.

**Weaknesses:**

* Existing literature (already cited in the paper, e.g. Zong et al.) has already shown that MLLMs in general do not benefit from support examples for VQA tasks. Consequently, what is presented as surprising in the paper is already known. In a way the paper discusses existing works that support the given conclusion, but at the same time it seems to present the conclusion as novel and surprising.
* The paper focuses only on VQA tasks for studying ICL, but this has been shown (in the cited literature) not to be an interesting ICL setup as MLLMs can solve such tasks well without the support examples. In fact, also the performances reported in the paper are at the level where most tasks are successfully solved. Family of tasks where ICL was shown beneficial in literature is mentioned in the paper but seemingly ignored for the analysis and evaluation. This means one cannot generalize the findings to ICL in general.
* The abstract seems to suggest that there is relatively significant focus on OOD analysis, but this section is in fact only a rather short one and gives mixed results.

**Questions:**

* Does adding reasoning rationales to ICL in non-VQA tasks help improve performance? How beneficial is it in general ICL? (comparison with and without reasoning rationals)

---

### Official Review · Reviewer_e1Df · 2025-11-01

**Soundness:** 2
**Presentation:** 3
**Contribution:** 2
**Rating:** 2
**Confidence:** 3

**Summary:**

This paper questions whether VLMs actually perform in-context learning or just pattern match. The authors test models under distribution shift and add reasoning rationales to demonstrations. Across many models (3B-72B) and datasets, they find VLMs show little sensitivity to shot count or demo quality, often performing worse with more examples. While the empirical work is extensive, the paper lacks novelty (similar findings in concurrent work), offers no solutions, and has methodological issues that undermine its claims. The format consistency finding is interesting but feels disconnected from the main narrative.

**Strengths:**

- The idea has a good motivation
- Well-written with good task categorization

**Weaknesses:**

- Core finding (VLMs use shallow heuristics, not true ICL) already documented in:
+ Folco Bertini Baldassini, Mustafa Shukor, Matthieu Cord, Laure Soulier, and Benjamin Piwowarski. "What makes multimodal in-context learning work?" In Proceedings of the IEEE/CVF Conference on Computer Vision and Pattern Recognition, pages 1539–1550, 2024.
+ Libo Qin, Qiguang Chen, Hao Fei, Zhi Chen, Min Li, and Wanxiang Che. "What factors affect multi-modal in-context learning? an in-depth exploration." arXiv preprint arXiv:2410.20482, 2024.
+ Shuo Chen, Zhen Han, Bailan He, Jianzhe Liu, Mark Buckley, Yao Qin, Philip Torr, Volker Tresp, and Jindong Gu. "Can Multimodal Large Language Models Truly Perform Multimodal In-Context Learning?" arXiv preprint arXiv:2311.18021, December 2024.
- The authors identify problems but proposes nothing to fix them
- Prefix matching designed for language-only; unclear if applicable to multimodal
- Heavy reliance on GPT-4o mini as judge introduces bias
- No comparison with instruction-tuned models for format handling

**Questions:**

Please answer the questions in the weaknesses section

---

### Official Review · Reviewer_4EtJ · 2025-11-04

**Soundness:** 2
**Presentation:** 3
**Contribution:** 2
**Rating:** 4
**Confidence:** 4

**Summary:**

This paper studies the Multimodal In-Context Learning (MMICL) ability of Vision-Language Models (VLMs), focusing on whether these models can effectively learn from demonstrations for downstream tasks. The paper identifies that MMICL performance drops as more out-of-distribution (OOD) demonstrations are provided. Experiments on reasoning VLMs show that including rationales in demonstrations improves MMICL performance. However, MMICL on reasoning VLMs still shows limited improvement compared to zero-shot inference. A mechanistic analysis reveals that the models exhibit weak prefix matching and lack induction-head-like behavior.

**Strengths:**

This paper studies an important direction for VLMs, and presents an analysis of reasoning VLMs.

The analysis and findings from the extensive experiments with various VLMs are likely to be of interest to the community.

**Weaknesses:**

The findings about performance degradation and pattern copying are not surprising for VLMs, as various previous works have pointed out this issue and published benchmarks to truly benchmark while avoiding the pattern copying issue (VLICL [1], TrueMICL [2], which can be better discussed in the paper).

Besides, the motivation of the whole analysis is unclear.

This paper classifies the MMICL tasks into two categories (Case I: Well Defined w/o Demos, such as OKVQA and Case II: Ill defined w/o demos, such as Operator Induction), and focuses more on Case I by analyzing each dataset's performance in MMICL.

What should the model learn from demos to improve MMICL performance, especially with OOD demos? What information should it extract from demonstrations? For instance, if humans receive these extra demos (OOD or ID), why and how would they perform better? What abilities improve with access to these demos? Without a detailed analysis of these questions, it's difficult to pinpoint concrete reasons for performance degradation, let alone improving it.

Moreover, regarding the retriever for reasoning VLMs: the retriever failed on reasoning-intensive datasets — what is the reason? Should we retrieve samples based on the desired rationale? Improving the MMICL via providing demos with consistent formats is also not a new message, as previous studies have identified the important role of label space and answer format.

[1] VL-ICL Bench: The Devil in the Details of Multimodal In-Context Learning, ICLR 2025
[2] Code of True Multimodal In-Context Learning Needs Attention to the Visual Context, COLM2025

**Questions:**

Please see weaknesses above.

---

### Note · Authors · 2025-11-15

I have read and agree with the venue's withdrawal policy on behalf of myself and my co-authors.